# Accurate and Efficient Intracranial Hemorrhage Detection and Subtype Classification in 3D CT Scans with Convolutional and Long Short-Term Memory Neural Networks

**DOI:** 10.3390/s20195611

**Published:** 2020-10-01

**Authors:** Mihail Burduja, Radu Tudor Ionescu, Nicolae Verga

**Affiliations:** 1Department of Computer Science, University of Bucharest, 14 Academiei, 010014 Bucharest, Romania; warchildmd@gmail.com; 2Romanian Young Academy, University of Bucharest, 90 Panduri, 050663 Bucharest, Romania; 3Department of Radiotherapy, Oncology and Hematology, “Carol Davila” University of Medicine and Pharmacy, 27 Dionisie Lupu, 020021 Bucharest, Romania

**Keywords:** intracranial hemorrhage detection, intracranial hemorrhage subtype classification, convolutional neural networks, long short-term memory networks

## Abstract

In this paper, we present our system for the RSNA Intracranial Hemorrhage Detection challenge, which is based on the RSNA 2019 Brain CT Hemorrhage dataset. The proposed system is based on a lightweight deep neural network architecture composed of a convolutional neural network (CNN) that takes as input individual CT slices, and a Long Short-Term Memory (LSTM) network that takes as input multiple feature embeddings provided by the CNN. For efficient processing, we consider various feature selection methods to produce a subset of useful CNN features for the LSTM. Furthermore, we reduce the CT slices by a factor of 2×, which enables us to train the model faster. Even if our model is designed to balance speed and accuracy, we report a weighted mean log loss of 0.04989 on the final test set, which places us in the top 30 ranking (2%) from a total of 1345 participants. While our computing infrastructure does not allow it, processing CT slices at their original scale is likely to improve performance. In order to enable others to reproduce our results, we provide our code as open source. After the challenge, we conducted a subjective intracranial hemorrhage detection assessment by radiologists, indicating that the performance of our deep model is on par with that of doctors specialized in reading CT scans. Another contribution of our work is to integrate Grad-CAM visualizations in our system, providing useful explanations for its predictions. We therefore consider our system as a viable option when a fast diagnosis or a second opinion on intracranial hemorrhage detection are needed.

## 1. Introduction

According to Urden et al. [1], intracranial hemorrhages account for roughly 10% of strokes in the United States. While hemorrhagic strokes are less frequent than ischemic strokes (87%), the former ones present a higher mortality rate. Indeed, it seems that between 37% and 38% of hemorrhagic strokes result in death within 30 days. With approximately 795,000 strokes per year in the United States alone, the number of yearly death cases caused by intracranial hemorrhage is in the range of 30,000. Therefore, intracranial hemorrhage is considered one of the most critical health conditions, demanding rapid intervention and intensive post-traumatic healthcare. Rapid intervention requires an urgent diagnosis for this life-threatening condition. Severe headache or loss of consciousness are neurological symptoms often associated with intracranial hemorrhage. When a patient shows such symptoms, highly trained radiologists typically analyze CT scans of the patient’s brain to find and determine the type of hemorrhage. However, the manual analysis performed by radiologists is complicated and usually time consuming, inherently and undesirably postponing the intervention. In these circumstances, the fast and automatic detection and classification of intracranial hemorrhage is of utter importance.

In this paper, we propose an accurate and efficient system based on a lightweight deep neural network architecture composed of a convolutional neural network (CNN) [2,3] that takes as input individual CT slices, and a Long Short-Term Memory (LSTM) network [4] that takes as input multiple feature embeddings (corresponding to an entire CT scan) provided by the CNN. To our knowledge, there are only a handful of works that employed a similar neural architecture, based on convolutional and recurrent layers, for intracranial hemorrhage detection [5,6,7,8]. Among these, there is only one prior work that classifies intracranial hemorrhage into subtypes [8]. The main novelty of our work, differentiating it from the prior literature, consists in integrating a feature selection method for more efficient processing. As such, we consider various ways of selecting useful features from the penultimate layer of the CNN. First of all, we select features by considering the weight assigned by the Softmax layer of the CNN to each feature, with respect to each subtype of intracranial hemorrhage. Second of all, we consider Principal Component Analysis (PCA) as an alternative approach for feature selection. The selected features are given as input to the LSTM, irrespective of the feature selection method. By taking into account slices from an entire CT scan, the LSTM mimics the review process performed by doctors, thus improving the slice-level predictions. In addition, another novelty of our framework is to feed the CNN predictions as input to the Softmax classification layer of the LSTM, concatenating them with the LSTM feature vectors. This brings an accuracy boost with negligible computational overhead. Furthermore, we downscale the CT slices by a factor of 2×, which allows us to train the model faster. Certainly, reducing the input size and performing feature selection also translates into faster inference times.

Motivated by the goal of providing fast diagnosis and constrained by our computing infrastructure, we designed a system that is focused on minimizing the computational time. Nevertheless, we present empirical results indicating that, using our system, we ranked in the top 2% (27th place) in the RSNA Intracranial Hemorrhage Detection challenge (https://www.kaggle.com/c/rsna-intracranial-hemorrhage-detection/), from a total of 1345 participants. In the challenge, the systems were ranked by the weighted mean log loss, and our best score was 0.04989. However, this score is not indicative of how well the system performs in comparison to highly trained radiologists. Therefore, after the challenge, we conducted a subjective intracranial hemorrhage detection assessment by radiologists, indicating that the classification accuracy rate of our deep model is on par with that of doctors specialized in reading CT scans.

In order to allow further developments and results replication, we provide our code as open source in a public repository (https://github.com/warchildmd/ihd). Another contribution of our work is to integrate Grad-CAM visualizations [9] in our system, providing useful explanations for its predictions. In this paper, we also provide interpretations by a team of radiologists for some of the visualizations produced for difficult or controversial cases. Considering the presented facts, we believe that our system is a viable option when a fast diagnosis or a second opinion on intracranial hemorrhage detection are needed.

In summary, we make four contributions:We propose a joint convolutional and recurrent neural model for intracranial hemorrhage detection and subtype classification, which incorporates a feature selection method that improves the computational efficiency of our model.We conduct a human evaluation study to compare the accuracy level of our system to that of highly trained doctors.We provide our code online for download, allowing our results to be easily replicated.We provide interpretations explaining the decisions of our model.

The rest of this paper is organized as follows. We present related work in Section 2. We describe our method in detail in Section 3. Our experiments and results are presented in Section 4. We provide interpretations and visualizations to explain our model’s decisions in Section 5. Finally, we draw our conclusions in Section 6.

## 2. Related Work

After the initial success of deep learning [10] in object recognition from images [3,11], deep neural networks have been adopted for a broad range of tasks in medical imaging, ranging from cell segmentation [12] and cancer detection [13,14,15,16,17] to intracranial hemorrhage detection [5,8,18,19,20,21,22] and CT/MRI super-resolution [23,24,25,26]. Since we address the task of intracranial hemorrhage detection, we consider related works that are focused on the same task as ours [5,6,7,8,18,19,20,21,22,27,28,29,30], as well as works that study intracranial hemorrhage segmentation [31,32,33,34,35,36,37,38,39,40,41,42,43,44].

While most of the recent works proposed deep learning approaches such as convolutional neural networks [18,20,21,22,27,29,30,37], fully-convolutional networks (FCNs) [19,32,33,36,38,39] and hybrid convolutional and recurrent models [5,6,7,8], there are still some recent works based on conventional machine learning methods, e.g., superpixels [43,44], fuzzy C-means [31,35], level set [42,43], histogram analysis [41], thresholding [40] and continuous max-flow [34].

Bhadauria et al. [31] employed fuzzy C-means and region-based active contour for brain hemorrhage segmentation. The authors automatically estimated the parameters that control the propagation of contour through fuzzy C-means clustering. Gautam et al. [35] used fuzzy C-means with a different purpose, that of removing the skull, applying wavelet transform and thresholding on the remaining tissue. Two other recent methods are based on the idea of thresholding [40,41]. Pszczolkowski et al. [40] used the mean and the median intensity in T2-weighted MRI to compute a segmentation threshold, while Ray et al. [41] analyzed the density of pixels at different levels of intensity to find the desired threshold. Some methods [43,44] partitioned the 2D or 3D images into small segments known as superpixels or supervoxels, respectively. Sun et al. [44] obtained supervoxels by using C-means clustering and by removing the smaller clusters. Soltaninejad et al. [43] applied their method on both 2D or 3D images, grouping the superpixels and supervoxels based on their average intensity. They determined the final contour of the hemorrhage region by a distance-regularized level set method. Another method based on the distance-regularized level set method was proposed by Shahangian et al. [42]. In addition to hemorrhage segmentation, Shahangian et al. [42] performed classification of segmented regions by extracting shape and texture features. Chung et al. [34] solved a convex optimization function using the continuous max-flow algorithm. They used an edge indicator for regularization. The methods based on statistical or standard machine learning approaches presented so far [31,34,35,40,41,42,43,44] are not suitable for large-scale datasets, requiring manual adjustment of parameters and careful monitoring of the results. Indeed, Chung et al. [34] acknowledge that such methods can only be used in a semi-automatic scenario. Unlike these methods, we employ deep neural networks, which exhibit a strong generalization capacity and are able to provide accurate and robust results on large image databases.

One of the first works to adopt deep learning for intracranial hemorrhage detection is that of Phong et al. [21]. The authors found that convolutional neural networks pre-trained on ImageNet [45] can successfully be used for intracranial hemorrhage diagnosis. Lee et al. [20] proposed a system composed of four different CNN models pre-trained on ImageNet, while also integrating a method to explain the decisions through Grad-CAM visualizations [9]. Different from Phong et al. [21] and Lee et al. [20], Arbabshirani et al. [18] trained CNNs from scratch on a large dataset of nearly 50 thousands images. Islam et al. [37] trained a VGG-16 model with atrous (dilated) convolutions on CT slices. The output corresponding to each CT slice is subsequently processed by a 3D conditional random field that provides the final result considering entire CT scans. Chang et al. [27] employed a modified Mask R-CNN architecture [46], in which the backbone is a hybrid 3D and 2D version of Feature Pyramid Networks [47]. Ker et al. [29] applied image thresholding to improve the results of a 3D CNN that takes as input multiple consecutive slices. As the 3D convolutions are rather computationally expensive, the authors adopted a shallow architecture with three convolutional and two fully-connected layers. Saab et al. [30] formulated intracranial hemorrhage detection as a multiple-instance learning task, in which the labels are provided at the scan level instead of the slice level. This allows them to incorporate data from a large number of patients. Rao et al. [22] used a deep CNN in a framework that finds misdiagnosis of intracranial hemorrhage in order to mitigate the impact of negative patient outcomes. Unlike the methods that employed CNNs for intracranial hemorrhage detection [18,20,21,22,27,29,30,37], with or without pre-training, we take our model a step further and use the CNN activations maps from the penultimate layer as input to an LSTM network that processes entire CT scans. This gives our model the ability to correct erratic predictions at the slice level.

For intracranial hemorrhage segmentation, there are a few works [32,36,38,39] that employed the U-Net model [12], a fully-convolutional network shaped like an auto-encoder with skip connections. Hssayeni et al. [36] used the standard U-Net model, their main contribution being a dataset of 82 CT scans. Cho et al. [32] combined the conventional U-Net architecture with an affinity graph to learn pixel connectivity and obtain smooth (less noisy) segmentation results. Kwon et al. [38] proposed a Siamese U-Net architecture, in which the long skip-connections are replaced by a Siam block. They reported better results than a conventional U-Net. In order to alleviate the class imbalance problem in their dataset, Patel et al. [39] introduced a weight map for training a U-Net model with 3D convolutions. Cho et al. [33] proposed a cascaded framework for intracranial hemorrhage detection and segmentation. For the detection task, they employed a cascade of two CNN models. The positively labeled examples are passed through two FCN models and the corresponding segmentation maps are summed up into a final and more accurate segmentation map. Kuo et al. [19] trained a patch-based fully convolutional neural network based on a ResNet architecture with 38 layers and dilated convolutions. The authors reported accuracy rates comparable to that of trained radiologists for acute intracranial hemorrhage detection. The state-of-the-art methods based on FCN models [19,32,33,36,38,39] are mainly focused on the segmentation task, whereas our focus is on the detection task. Some of the FCN models are based on 3D convolutions, obtaining better segmentations by considering adjacent CT slices. For the detection task, we propose a more efficient architecture that processes 3D CT scans without requiring 3D convolutions. While being more computationally efficient, our CNN and LSTM architecture is not suitable for segmentation, although we can interpret the Grad-CAM visualizations as some sort of rough segmentation maps.

One of the first works to propose a joint CNN and LSTM model for intracranial hemorrhage detection is that of Grewal et al. [5]. Their method performs segmentation at multiple levels of granularity and binary classification of intracranial hemorrhage. Another method based on a similar architecture is proposed by Patel et al. [6]. The authors trained the model on a larger dataset of over 25 thousand slices. Vidya et al. [7] employed a CNN on CT sub-volumes, combining the outputs at the sub-volume level into an output at the scan level through a recurrent neural network (RNN). Their method is designed to identify hemorrhage, without being able to classify it into the corresponding subtype. Unlike these related methods [5,6,7], we propose a method that is able to classify intracranial hemorrhage into subtypes: epidural, intraparenchymal, intraventricular, subarachnoid, subdural. To our knowledge, the only method with a similar design and the same capability of detecting the five subtypes of intracranial hemorrhage is that of Ye et al. [8]. They employed a cascaded model, with an RNN for intracranial hemorrhage detection and another RNN for subtype classification. We propose a more efficient model that detects intracranial hemorrhage and classifies it into the corresponding subtype in a single shot. Setting our focus on efficiency and novelty, we employ a feature selection method in order to find the most useful features learned by our CNN, obtaining a compact representation to be used as input to our bidirectional LSTM. This aspect differentiates our model from the related literature.

## 3. Materials and Methods

Our framework is formed of a convolutional neural network, a feature selection method and a recurrent neural network, the training being divided into three stages. In the first stage, we train (fine-tune) our convolutional neural network for predicting intracranial hemorrhage types at the CT slice level. For the slice-level classification task, we start from a ResNeXt-101 32×8 d architecture [48] that was pre-trained using a large-scale image dataset unrelated to CT imaging [49]. Our next training stage consists in selecting useful features from the penultimate layer of the CNN. As feature selection method, we consider using either PCA or the weights assigned by the Softmax classification layer already included in the CNN. The resulting features are provided as input to the recurrent neural network. In the third and last training stage, we train our recurrent neural network for predicting intracranial hemorrhage types at the CT scan level. For the scan-level classification task, we employ a bidirectional LSTM, training it from scratch. By considering the CT slices from a CT scan all at once, the proposed model is able to improve its predictive capacity, eliminating inconsistent annotations at the slice level. Since doctors also look at multiple neighboring slices when providing their diagnostic, it seems natural to design a machine learning model based on the same principle. Indeed, our experiments indicate that considering information form multiple neighboring slices provides a boost in performance with respect to all the metrics considered in our evaluation. We further describe in detail the proposed joint CNN and LSTM architecture, loss function, data augmentation procedure and feature selection methods.

### 3.1. Data Preprocessing and Augmentation

The common format for storing and transmitting medical CT images is DICOM^®^ (Digital Imaging and Communications in Medicine), which contains metadata alongside the pixel data. The RSNA dataset [28] contains 16-bit images, meaning that each image is represented on a scale with 65,536 levels of gray. The value of each pixel represents the density of the tissue measured in Hounsfield units (HU). Since most monitors can only display 256 levels of gray (8-bit images), specialized DICOM software allows radiologists to focus on different intensity windows of HU, which are used to study different tissue types and various pathologies. An intensity window is commonly defined through its center (WC) and its width (WW), specifying a linear conversion from HU to pixels values to be displayed. Radiologists usually look at predefined intensity windows in order to detect a specific pathology. Following the recommendation of trained radiologists for intracranial hemorrhage detection, we consider three HU windows, each focusing on different types of tissue: brain window (WC=40, WW=80), subdural window (WC=80, WW=200) and soft tissue window (WC=40, WW=380). By applying a single HU window on a CT slice, we obtain a grayscale (8-bit) image. The images resulted after applying the three intensity windows recommended by radiologists are combined into a single three-channel image, as shown in Figure 1. While the original CT slices are formed of 512×512 pixels, we reduce their spatial dimension to cope with the large training set of over 700 thousand slices [28]. The CT slices are reduced to 256×256 pixels using linear interpolation. Hence, the input of our neural network is 256×256×3. Following the standard preprocessing procedure in training CNN models [50], we normalize all the resulting images by subtracting the mean ([0.1738,0.1433,0.1970]) and dividing them by the standard deviation ([0.3161,0.2850,0.3111]) with respect to each channel.

We applied some transformations to augment the dataset, generating multiple variations of the images in order to increase our model’s generalization capacity. Using the Albumentations library [51], the transformations that we chosen for data augmentation are: horizontal flipping, shifting, rotation, scaling and brightness adjustment. Some images before and after data augmentation are shown in Figure 2.

### 3.2. Model and Feature Selection

Convolutional neural networks [2,3] are a type of deep neural networks mainly used in computer vision. The popularity of CNN models has grown lately due the availability of large annotated datasets and the advances in GPU computing, allowing the efficient training of neural models on large datasets. The main advantage of deep models over previously-used methods for solving computer vision tasks is represented by that fact that deep models do no require domain experts to handcraft useful features, automatically learning to extract interesting features through end-to-end training. The main building blocks of CNN models are the convolutional layers, which are typically organized sequentially. Convolutional layers are formed of filters (kernels) that learn to detect various features, producing activation maps after applying the convolution operation on their input. The layers closer to the input image usually learn low-level features, e.g., edges, bars, corners or stains. Going at deeper convolutional layers, we typically observe more complex features, e.g., texture patterns, object parts, which are obtained by convolving lower-level features. Convolutional neural networks achieved state-of-the-art results on a broad range of computer vision [3,11,50,52,53,54,55] and medical imaging [12,19,24] tasks. In many cases, their success is in large part due to the availability of pre-trained models on large-scale datasets, which are highly transferable to other tasks, requiring only some fine-tuning. For instance, there are many works [20,21,56,57,58,59] tackling medical imaging tasks with models pre-trained on ImageNet [45]. In fact, Raghu et al. [58] stated that “transfer learning from natural image datasets using standard large models and corresponding pre-trained weights has become a de-facto method for deep learning applications to medical imaging”. Following this de-facto standard in medical imaging, we started with a recent pre-trained CNN model introduced by Xie et al. [48], namely a ResNeXt-101 32×8 d. This model was trained by Xie et al. [48] on 940 million public images [49].

The ResNeXt architecture of Xie et al. [48] is a highly modularized network architecture for image classification inspired by VGG [50], Inception [54] and ResNet [11]. ResNeXt is comprised of residual blocks that share the same topology with one another. The design that we employed, ResNeXt-101 32×8 d, is similar to ResNet-101 [11], which is comprised of 101 layers. As the name implies, the chosen architecture has a cardinality of 32 and a bottleneck width of 8. Xie et al. [48] define cardinality as the size of the set of transformations (alternative neural pathways), which seems to play a more important role than the width or the depth of the neural network. Each ResNeXt block contains three convolutional layers, with the first and the last layers having filters with a spatial support of 1×1 and the second one having filters with a spatial support of 3×3, respectively. The number of filters in the second convolutional layer is typically much smaller, hence, the layer acts as a bottleneck layer. The bottleneck width refers to the number of filters in the bottleneck layer. We refer the reader to Xie et al. [48] for additional details on the concepts of cardinality and bottleneck width. The number of learnable parameters is approximately 88 million, generating a complexity of 16B FLOPS. The ResNeXt is pre-trained in a weakly supervised manner, as explained by Mahajan et al. [49]. We feed the chosen CNN with singular CT slices in order to fine-tune it on the task of predicting the presence and the type of intracranial hemorrhage per CT slice. To fine-tune the network for our task, we removed the pre-trained classification layer and replaced it with our own Softmax classification layer, which is initialized with random weights.

Long Short-Term Memory networks [4] are a type of recurrent neural networks that are suited for sequences of data points. LSTM networks have produced notable results in language modeling, machine translation and other problems that rely on sequences as input. In CT brain hemorrhage detection, we can interpret a CT scan as a sequence of CT slices, that have a natural (spatial) order. We note that taking into account neighboring slices can provide a performance boost to the model, as shown in the experiments. The LSTM does not need a fixed number of slices as input, i.e., CT scans with various numbers of slices are automatically handled by the model.

Combining the CNN with the LSTM is usually done by extracting features from the CT slices using the CNN and then feeding the features to the LSTM, thus taking into account the CT scan as whole. Our complete architecture is illustrated in Figure 3. The ResNeXt-101 32×8 d produces a 2048-dimensional feature vector for each input CT slice. As a CT scan can contain tens of slices, the input size for the LSTM, which receives all the slices of a CT scan at once, could rapidly reach more than 100 thousand components, e.g., 50×2048= 102,400. In order to reduce the training time and the need for computing power for the LSTM, we decided to apply a feature selection method to reduce the 2048-dimensional vectors to a manageable size. We tried two different methods of feature selection. Our first method is based selecting features with the lowest or highest weights (in absolute value) for each output class, the considered weights being those learned by the Softmax classification layer of ResNeXt-101 32×8 d. The features with higher weights should be more informative for the intracranial hemorrhage classes. On the other hand, we observed that these features have very low variance. Hence, we also considered selecting the features with the lowest weights. As an alternative approach for feature selection, we considered PCA, a dimensionality reduction method that projects the data points in a new subspace, in which the features (components) are orthogonal and in decreasing order of variance. In the experiments, we applied PCA on a subset of 30,000 feature vectors. The number of components (120 or 192) was chosen to optimize the size of the output embeddings, while explaining as much of the variance in data as possible.

As a CT scan contains multiple slices, reducing the false positive and false negative rates is possible by training on multiple slices of the same scan together as input, e.g., a spurious detection in one slice can be corrected by considering neighboring CT slices. The number of slices in a CT scan varies, but we opted for a model that can adapt to different numbers of slices as input. More precisely, we employed an LSTM model that takes as input a sequence of CNN embeddings and outputs 256-dimensional LSTM embeddings. Our recurrent model is a 3-layer bidirectional LSTM with a dropout rate of 0.3 and a cell state vector of 256 components. As a final attempt to boost the performance of our model, we concatenated the class probabilities of the ResNeXt-101 32×8 d model to the 256-dimensional LSTM feature vectors that go into the Softmax classification layer of the bidirectional LSTM, as illustrated in Figure 3.

Both CNN and LSTM models perform two classification tasks at once. The first classification task is to classify whether the CT scan/slice contains an intracranial hemorrhage (this is referred to intracranial hemorrhage detection), while the second task is to classify the type of hemorrhage into five classes (this is referred to intracranial hemorrhage subtype classification).

### 3.3. Loss and Optimization

Intracranial hemorrhage detection and subtype classification can be considered jointly as a multi-label classification problem. This means that one data sample, e.g., a CT slice or a CT scan, can have multiple types of hemorrhage present in the same time. We train our models jointly on detection and 5-way subtype classification. Our models output values between 0 and 1 indicating the probability of occurrence for each type of intracranial hemorrhage as well as the probability of having intracranial hemorrhage (any subtype). In total, a neural model contains six units on the last (classification) layer, giving as output six independent class probabilities. We use the multi-label log loss, summing the binary cross-entropy (BCE) loss for each type of hemorrhage for a given input sample:(1)$$L_{multi-BCE}\left(y,\hat{y}\right)=-\sum_{t=1}^{6}y_{t}\cdot log\left({\hat{y}}_{t}\right)+\left(1-y_{t}\right)\cdot log\left(1-{\hat{y}}_{t}\right),$$
where $y_{t}\in \left\{0,1\right\}$ represents the ground-truth label for a class *t* and ${\hat{y}}_{t}\in \left[0,1\right]$ represents the prediction (class probability) for class *t*. There are 5 subtype classes (epidural, intraparenchymal, intraventricular, subarachnoid and subdural) and one generic class for intracranial hemorrhage, so t∈{1,2,...,6}.

The main factor contributing to choosing the multi-label log loss function was that systems competing in the RSNA Intracranial Hemorrhage Detection challenge were ranked by using a weighted average version of this loss as the performance metric. The best way to achieve a good performance for a specific metric is to optimize directly for the respective metric.

Our CNN and LSTM models were trained separately, using the Adam optimizer [60] for both models, but with different learning rates. We trained (fine-tuned) the CNN for three epochs, using a learning rate of $10^{-4}$ for the first two epochs and a learning rate of $2\cdot 10^{-5}$ for the third epoch. We trained the LSTM for four epochs using a learning rate of $10^{-4}$.

## 4. Experiments and Results

### 4.1. Data Set

The Radiological Society of North America (RSNA) 2019 Brain CT Hemorrhage dataset [28] was built from scratch for the 2019 RSNA Intracranial Hemorrhage Detection challenge held on Kaggle (https://www.kaggle.com/c/rsna-intracranial-hemorrhage-detection/). The dataset is comprised of CT scans from three institutions: Stanford University, Universidade Federal de Sao Paulo and Thomas Jeffereson University Hospital. It contains 25,272 CT examinations (scans) with a total of 870,301 CT slices. Each CT scan was labeled by annotators, using five brain hemorrhage types as labels: epidural (EPH), intraparenchymal (IPH), intraventricular (IVH), subarachnoid (SAH), subdural (SDH). If a slice contains at least one type of intracranial hemorrhage, it is automatically labeled as having intracranial hemorrhage (ICH). The annotators did not have information on medical history, acuity of symptoms, age of the patient or previous examinations.

The dataset is split into a training set of 752,803 slices and a test set of 121,232 slices, as shown in Table 1. We further split the official training data into a training set of 728,513 slices and a validation set 24,290 slices. While the slices are provided independently, we note that the slices can be grouped by the scan ID, which is provided in the DICOM metadata. On average, each CT scan contains about 35 slices. In total, the training set contains 21,000 CT scans, the validation set contains 744 CT scans and the test set contains 3528 CT scans. As illustrated in Figure 4, the class distribution in the dataset is not balanced, the EPH type being underrepresented. We note that the test labels are not known, the evaluation on the test set being possible only by submitting predictions through Kaggle.

### 4.2. Experimental Setup

#### 4.2.1. Implementation Details

We trained our models using a single NVIDIA P100 GPU provided on the Kaggle platform, using NVIDIA Apex for Automatic Mixed Precision (AMP). All images were downsampled to 256×256 pixels, allowing us to train on larger mini-batches and to reduce the training time. We used mini-batches of 32 samples for training the CNN model, and mini-batches of slices from an entire CT scan (with a variable amount of slices) for training the bidirectional LSTM. The resulting CNN embeddings have 2048 features, which were reduced in numbers by selecting the top 120 (or top 192) features using a feature selection method. The code was implemented in Pytorch, using the Adam optimizer [60] with multi-label mean log loss.

#### 4.2.2. Evaluation Metrics

We employed multiple evaluation metrics for assessing the performance levels of our models, both at the slice level and at the whole scan level. First of all, we used the multi-label weighted mean log loss as an evaluation metric at the slice level, which is the same metric used to rank participants in the RSNA Intracranial Hemorrhage Detection challenge. While this is the official metric, it does not provide any useful insights about our model. We therefore include a set of more informative metrics, namely the accuracy, the area under the ROC (Receiver Operating Characteristics) curve (AUC), the sensitivity and the specificity. Since we do not have access to the test labels, we reported the values for the whole set of metrics only on the validation set.

### 4.3. Results on the Official Test Set

We present results with various preliminary versions of our framework, emphasizing the effect of various components over the overall performance. We experimented with different CNN and LSTM architectures, as well as various feature selection methods. The corresponding results obtained on the official test set of the RSNA Intracranial Hemorrhage Detection challenge are presented in Table 2.

As our CNN model, we considered two options, namely ResNeXt-101 32×8 d [48] and EfficientNet-B4 [61]. From the range of EfficientNet architectures proposed by Tan et al. [61], we opted for the largest architecture that we could afford to train on our dataset, which is EfficientNet-B4. After fine-tuning both ResNeXt-101 32×8 d and EfficientNet-B4 on the training set, we observed that ResNeXt-101 32×8 d attains a much lower log loss (0.06259). We therefore opted for this CNN model in the subsequent experiments.

As previously mentioned, the CNN does not take into account the neighboring slices, making predictions on a slice by slice basis. In order to have a model that considers all slices in a CT scan at once, we introduced a bidirectional LSTM in our framework. We initially trained the bidirectional LSTM on the class probabilities provided by the ResNeXt-101 32×8 d model, attaining a log loss of 0.05540. The LSTM would benefit from having more values in the input vectors, which leads to the idea of using the 2048-dimensional features from the penultimate layer of the ResNeXt-101 32×8 d model. However, we were not able to perform this experiment, being constrained by the computing infrastructure provided by Kaggle. A good alternative, however, is to apply a feature selection method first, reducing the dimensionality of the feature vectors from 2048 to something more manageable. Our first method of feature selection was to consider the features with the largest standard deviation, resulting in a log loss of 0.05212. Another option to select features was to consider the weights from the Softmax layer of the ResNeXt-101 32×8 d model. We tried using the top 192 features for the bidirectional LSTM, selecting the features corresponding to either the largest or the smallest weights in absolute value. Interestingly, we obtained a lower log loss with the features having the smallest absolute values. The third feature selection method that we tried in the experiments is PCA. Using the same bidirectional LSTM architecture as before and the same number of features (192), we obtained a log loss of 0.05096. Shallower or slimmer bidirectional LSTM architectures provide slightly worse results. From the considered feature selection methods, we opted for PCA for our final submissions. Another attempt was to reduce the dimensionality of the CNN features vectors even more. We obtained an improved log loss of 0.05022 with 120-dimensional CNN features vectors, suggesting that a more aggressive feature selection is useful. In order to confirm this hypothesis on another case, we applied it for the feature selection method based on the weights having smallest absolute values, reaching a log loss of 0.05035, down from 0.05136. We thus conclude that selecting the top 120 features is more effective that selecting the top 192 features. While this does not imply that our performance would be worse without feature selection (we were not able to perform experiments without feature selection on our computing infrastructure), it surely points in this direction. To gain some performance boost, we tried to feed the predictions of the ResNeXt-101 32×8 d as input to the Softmax classification layer of the bidirectional LSTM, concatenating the class probabilities to the 256-dimensional feature vectors resulted from the penultimate layer of the bidirectional LSTM. This furthered lowered our log loss score down to 0.04989. With this score, we rank on 27th place out of 1345 participants.

While we were not able to train our framework without feature selection on images of 512×512 pixels until convergence, we trained it for a couple of iterations, comparing its computational time with respect to our final model, which relies on PCA to reduce the feature vectors to 120 dimensions, taking images of 256×256 pixels as input. The model trained on full-resolution images requires 128 milliseconds per image, while our model requires only 36 milliseconds for one image. For one epoch over the training set (728,513 slices), the model trained on full-resolution images requires about 26 h. Our model passes through the entire training set in only 7 h, staying under the maximum of 9 h for one run on the Kaggle platform. This is the main constraint that forced us to design a lighter neural framework.

### 4.4. Results on the Validation Set

In Table 3, we present additional metrics for our best model versus the plain ResNeXt-101 32×8 d on the validation set. Our accuracy rates and ROC AUC scores at the slice level are typically above 96.00%. We generally observe that our scores for these metrics are lower at the scan level than at the slice level. For the EPH subtype, we obtain the lowest sensitivity at the slice level (44.30%) and at the scan level (40.00%), most likely because this subtype is underrepresented in the dataset (see Figure 4). Our highest sensitivity at both slice and scan levels is attained on the IVH subtype, the values being slightly above 91%. Our model seems to produce very high specificity scores, surpassing the 99% threshold in many cases. We note that the main advantage of using the bidirectional LSTM is represented by the significant performance gains in terms of sensitivity at the slice level. Indeed, for EPH, IPH and SAH subtypes, the improvements brought by the bidirectional LSTM are higher than 10%.

## 5. Assessment by Radiologists and Discussion

While our best system attains an impressive log loss value of 0.04989, placing us in the top 30 ranking from a total of 1345 participants in the RSNA Intracranial Hemorrhage Detection challenge, it is hard to assess how well it performs with respect to trained radiologists, without a direct comparison. Therefore, we decided to use 100 CT scans (24,290 slices), held out from the validation set, to compare our deep learning models with three specialists willing to participate in our study. Some statistics regarding the ground-truth labels for the selected scans are provided in Table 4. We note that some CT scans can have multiple subtypes of ICH. Hence, the sum of positive labels for all subtypes (29+16+25+32=102) is higher than the number of scans labeled as ICH (52).

We provide the outcome of the subjective assessment study by radiologists on the held out CT scans in Table 5. We compare the annotations provided by three doctors with the labels provided by the CNN model and the joint CNN and LSTM model, respectively. The evaluation metrics are computed at the scan level. For a fair comparison with our systems, the doctors were not given access to the ground-truth labels during the annotation. We note that two of the doctors, doctor #2 and #3, are highly trained specialists, treating intracranial hemorrhage cases on a daily basis in their medical practice. This explains why their scores are much higher than the scores attained by doctor #1, who is a specialist in treating tumors through radiation therapy, rarely encountering intracranial hemorrhage cases in the day to day practice. The study reveals that our deep learning models attain scores very close to those of doctor #2 and #3, even surpassing them for the EPH subtype (our top accuracy being 100%) and the IPH subtype (our top accuracy being 92%). Our models are also better at generic ICH detection, our top accuracy being 98% with the plain CNN model. Based on our subjective assessment study by radiologists, we concluded with the doctors that our models exhibit sufficiently high accuracy levels to be used as a second opinion in their daily medical practice. The team of doctors considered the fact that our AI-based system attains performance levels comparable to those of trained radiologists very impressive.

In order to better assess the behavior of our system with respect to the radiologists included our study, we produced Grad-CAM visualizations [9] for the CT scans that were wrongly labeled by doctor #2 and by our CNN model (ResNeXt-101 32×8 d). We considered all scans that had at least one wrong label. There are 39 wrong labels given by doctor #2 and 40 wrong labels by our system. Interestingly, there is a high overlap (over 50%) between the wrong labels of doctor #2 and those of our system. Considering that there are 22 mistakes in common, we have strong reasons to suspect that the ground-truth labels might actually be wrong. As the ground-truth labels are also given by specialists [28], this is a likely explanation for the high overlap between the mistakes of doctor #2 and those of our CNN model. After seeing the ground-truth labels and making a careful reassessment, our team of doctors found at least 25 ground-truth labels that are wrong and another 5 that are disputable. Since the total number of labels for the 100 CT scans is 600 (6 classes × 100 scans), the wrong labels identified in the ground-truth account for less than 5% from the total amount, which is acceptable. Nevertheless, we emphasize that the CT scans selected for the Grad-CAM-based analysis are either difficult (a doctor or our system produced a wrong label) or controversial (the ground-truth labels are suspect of being wrong). In total, we gathered a set of 35 controversial or difficult scans, with a total of 414 controversial or difficult slices (we did not include slices without wrong labels). For each CT slice and for each positive label provided by our system, we produced a Grad-CAM visualization. Our team of doctors analyzed each and every Grad-CAM visualization, labeling each visualization as correct (the system focuses on the correct region, covering most of hemorrhage, without focusing on other regions), partially correct (the system focuses on some hemorrhage region while also focusing on non-hemorrhage regions or missing some other hemorrhage region) or incorrect (the hemorrhage region is completely outside the focus area of our system). From the total of 414 visualizations, 54% were labeled as correct, 19% as partially correct and the remaining 27% as incorrect. Considering that the 414 slices represent difficult or controversial cases, we believe that the percentage of incorrect visualizations is not high enough to represent a serious concern.

For the correct and partially correct visualizations, the doctors observed that the focus region of the system is not perfectly aligned with the hemorrhage area. When the CT slices contain SAH, a general observation is that our CNN-based system seems to concentrate on the eye orbits, the sinus or the nasal cavities, which is not right. Another general observation is that, for some visualizations labeled as incorrect, the system actually focuses on blood hemorrhage outside the skull, which is not entirely a bad thing, i.e., it is good that the system detects hemorrhage anywhere in the CT image, but it should learn to look for hemorrhage inside the skull only. Doctor #2 labeled 3 CT scans as EPH, but these were false positives. After seeing the ground-truth labels and the Grad-CAM visualizations, the team of doctors (which includes doctor #2) was able to correct the false positives for EPH, confirming the ground-truth labels which indicate that the respective scans contain SDH. The doctors observed that most of the correct visualizations are produced for specific hemorrhage types, namely IPH, IVH and SAH. The worst visualizations are typically produced for the generic ICH class (any type). This is because, during training, the system is not forced to detect all hemorrhage regions in order to label a CT slice as ICH. While it is sufficient for our system to detect at least one hemorrhage region to label the corresponding slice as ICH, the doctors would have expected the system to indicate all hemorrhage regions through the visualization. In order to visualize all hemorrhage regions, the doctors concluded that it is mandatory to jointly consider the Grad-CAM visualizations for all detected types of hemorrhage.

In order to better understand how doctors classified the Grad-CAM visualizations as correct, partially correct or incorrect, we next present a detailed analysis of five particular Grad-CAM examples from each category. In Figure 5, we illustrate a set of Grad-CAM visualizations that were labeled as correct by the doctors. In the first (left-most) sample, the system concentrates on the subdural hemorrhage in the left frontal lobe. In the second sample, the system correctly concentrates its attention on two hemorrhage regions, one in the left frontal lobe and one in the left parietal lobe. In the following two samples, we observe that our system focuses on hemorrhage in various regions, namely in the posterior median (tentorial) lobe (third example) and in the temporal right lobe (fourth example). In the fifth (right-most) sample in Figure 5, the hemorrhage within the focus area is located in the right paramedian region, the slice being labeled as IPH and IVH.

In Figure 6, we show a set of Grad-CAM visualizations that were labeled as partially correct by the doctors. In the first partially correct sample, the system focuses on the hemorrhage situated in the left frontal lobe, failing to indicate the hemorrhage in the left posterior temporalis. In the second sample from the left, the system focuses on the left frontal hemorrhage, while the left parietal (posterior) hemorrhage is not sufficiently highlighted. In the third Grad-CAM visualization, the system concentrates mostly on the tentorial left (posterior latero-median) hemorrhage, and only barely on the right temporal hemorrhage. The system seems to ignore the hemorrhage in the left temporal lobe. In the fourth partially correct example, the focus region includes the left median temporal IPH and the anterior left frontal SDH, but the right temporal SAH is not sufficiently highlighted by our system. In the fifth (right-most) sample in Figure 6, our system correctly annotates the SDH in the left frontal lobe, but the SDH in the parietal right lobe is not highlighted by the system.

Finally, we illustrate a set of incorrect Grad-CAM visualizations in Figure 7. In the left-most sample, there is a bilateral posterior hemorrhage that is outside the focus area of the system, which seems to have turned its attention on extracranial hemorrhage instead. In the second incorrect example, the system focuses on the left frontal lobe, but the lesion is located in the posterior median parietal lobe. In the third example, there is no intracranial injury, and the system seems to focus mostly on the extracranial area, which is not of interest. In the fourth incorrect visualization, there is a left temporal intraparenchymal lesion outside the focus area of the system, which seems to concentrate on the anterior median region instead. In the fifth example in Figure 7, there is an anterior left temporal lesion which the system does not concentrate on. Instead, the system system focuses on the anterior and the right paramedian regions, and also on the left anterior facial and extracranial mass.

As a generic conclusion of this analysis, the doctors found the Grad-CAM visualizations very useful, considering that the best application of our system is to provide a second informative opinion for ER patients that are suspected of ICH. Here, the system could aid ER doctors in making a fast and correct diagnostic, which could prove crucial in saving a patient’s life.

## 6. Conclusions

In this paper, we presented an approach based on convolutional and LSTM neural networks for intracranial hemorrhage detection in 3D CT scans. Our method is able to attain performance levels comparable to those of trained specialists, in the same time reaching a loss of 0.04989 in the RSNA Intracranial Hemorrhage Detection challenge, which brings us on the 27th place (top 2%) from a total of 1345 participants. All these results were obtained while making important design choices to reduce the computational footprint of our model, namely (i) the reduction of CT slices from 512×512 to 256×256 pixels and (ii) the inclusion of a feature selection method before passing the CNN embeddings as input to the bidirectional LSTM. We conclude that analyzing hemorrhage at the CT scan level, through a joint ResNeXt and BiLSTM network, is more effective than performing the analysis at the CT slice level, through a single ResNeXt model. We also deduce that performing feature selection is both efficient and effective when it comes to intracranial hemorrhage detection and subtype classification. Furthermore, we performed an analysis of Grad-CAM visualizations for some of the most difficult or controversial CT slices. The team of doctors that performed the manual annotations of the CT slices selected for their subjective assessment concluded that our system is an useful tool for their daily medical practice, providing useful predictions and visualizations.

In future work, we will work closely with the medical team to improve our model by incorporating their feedback. One of the most important problems is that our system can predict the correct label for a certain subtype of intracranial hemorrhage without considering all the locations for the respective subtype. While this behavior produces very good accuracy rates, it is not optimal for visualizing all the regions with hemorrhage. We aim to solve this issue in future work by considering pixel-level annotations, e.g., segments or bounding-boxes, allowing us to penalize the model during training when it fails to detect a certain hemorrhage region.

## Figures and Tables

**Figure 1 sensors-20-05611-f001:**
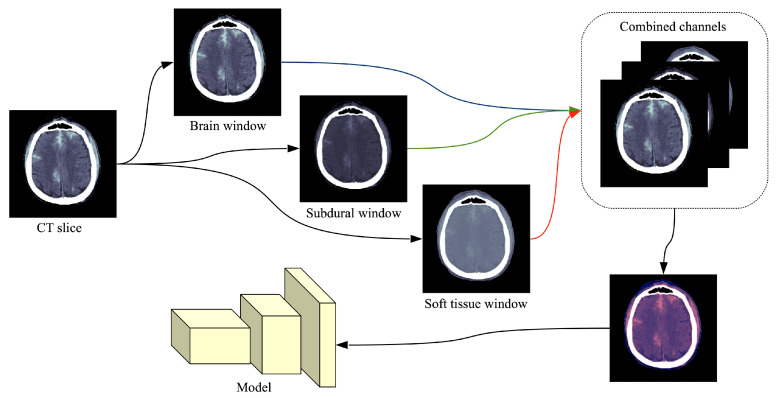
Preprocessing flow of a single CT slice in DICOM format. Each DICOM file is processed by extracting three different intensity windows (brain window, subdural window, soft tissue window), which are treated as three different channels. The resulting image is what the neural model takes as input. In order to illustrate the final image as an RGB image, we used the following correspondence: brain → red, subdural → blue, soft tissue → green. Best viewed in color.

**Figure 2 sensors-20-05611-f002:**
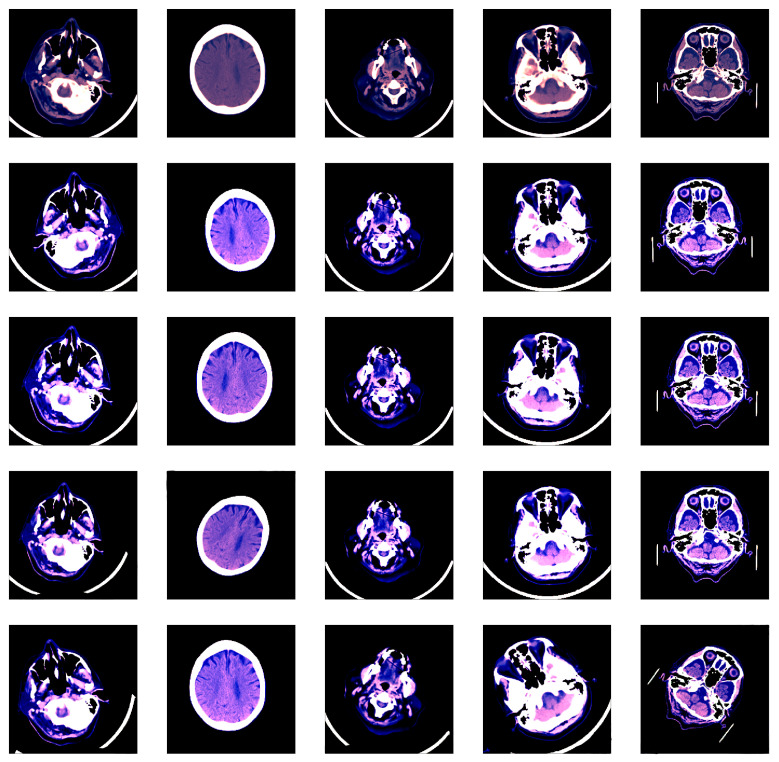
Various image transformations (rotation, scaling, shifting, brightness changes) used for data augmentation. Each column contains random transformations applied to the corresponding CT slice displayed on the first row. Best viewed in color.

**Figure 3 sensors-20-05611-f003:**
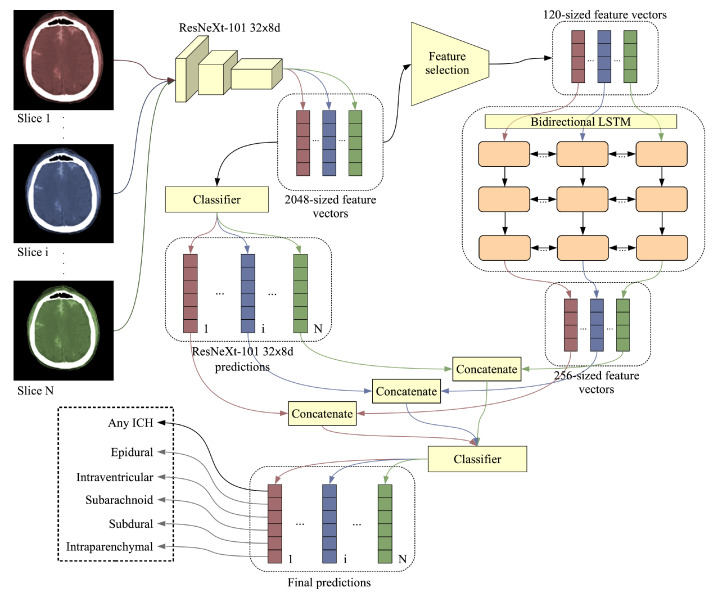
Proposed model architecture comprised of a convolutional neural network (CNN) model (ResNeXt 32×8 d), a feature selection method and a bidirectional Long Short-Term Memory (LSTM). The slices from a CT scan are fed into the CNN, which outputs 2048-dimensional feature vectors and 6-way class probabilities (5 types of intracranial hemorrhage and 1 for hemorrhage presence). A feature selection method is applied to reduce the CNN feature vectors to 120 dimensions. The resulting feature vectors are fed into a bidirectional LSTM network that outputs 256-dimensional feature vectors. A Softmax classifier takes the 6-way class probabilities of the CNN and the 256-dimensional feature vectors of the bidirectional LSTM as input, providing the final 6-way class probabilities. The ResNeXt model treats the slices independently, while the BiLSTM treats the slices jointly (an input example for the BiLSTM is a whole CT scan, which contains multiple slices). Best viewed in color.

**Figure 4 sensors-20-05611-f004:**
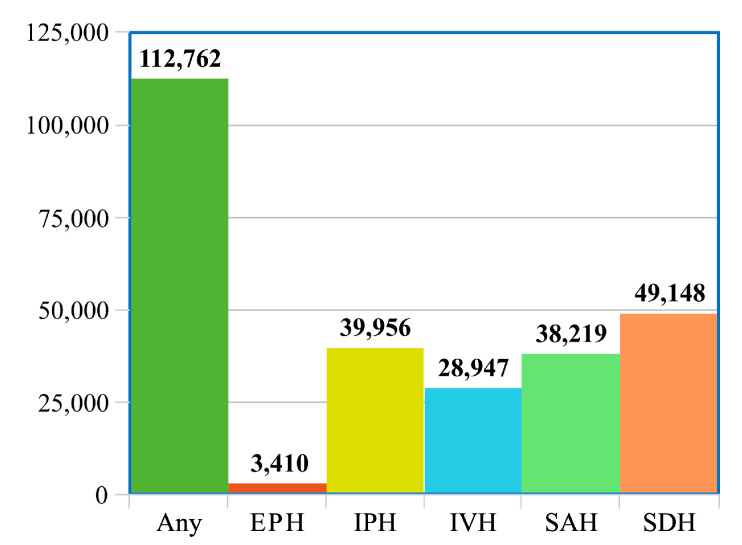
Distribution of hemorrhage types in the official training set, which is selected using stratified sampling from the RSNA 2019 Brain CT Hemorrhage dataset. Best viewed in color.

**Figure 5 sensors-20-05611-f005:**
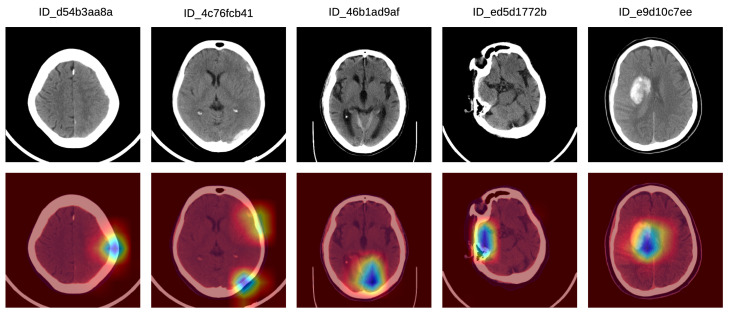
A selection of Grad-CAM visualizations that are labeled as correct by our team of doctors. Best viewed in color.

**Figure 6 sensors-20-05611-f006:**
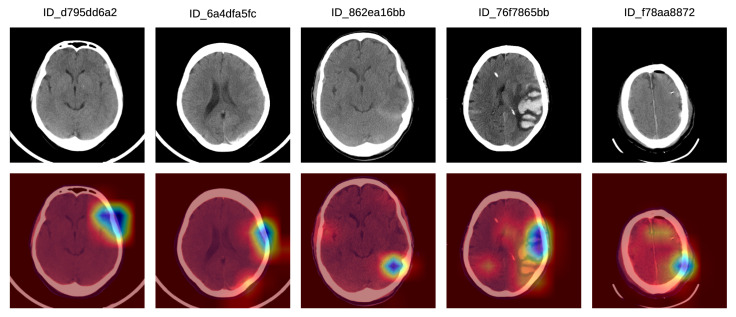
A selection of Grad-CAM visualizations that are labeled as partially correct by our team of doctors. Best viewed in color.

**Figure 7 sensors-20-05611-f007:**
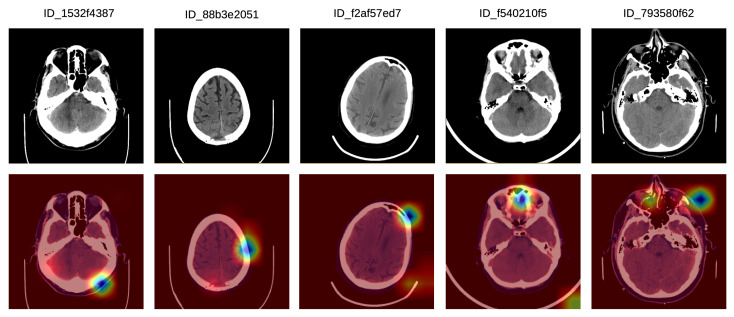
A selection of Grad-CAM visualizations that are labeled as incorrect by our team of doctors. Best viewed in color.

**Table 1 sensors-20-05611-t001:** Statistics regarding the number of scans and number slices for each hemorrhage type in the RSNA 2019 Brain CT Hemorrhage dataset. Our validation set is a subset of the official training set. The labels for the test set are unknown to participants.

Subtype	Training Set		Validation Set		Test Set	
	Slices	Scans	Slices	Scans	Slices	Scans
Any	104,199	8563	3734	319	-	-
Epidural	3066	344	79	10	-	-
Intraparenchymal	34,826	5130	1292	191	-	-
Intraventricular	25,381	3566	824	126	-	-
Subarachnoid	34,431	3788	1244	144	-	-
Subdural	45,479	3669	1687	143	-	-
None	624,314	12,437	20,556	425	-	-
Total	728,513	21,000	24,290	744	121,232	3528

**Table 2 sensors-20-05611-t002:** Multi-label weighted mean log loss on the official test set of the RSNA Intracranial Hemorrhage Detection challenge for different deep learning models. Results are reported for different CNN and LSTM architectures with and without feature selection. The Kaggle rank of each submission is also reported here. The best results are highlighted in bold.

Method	BiLSTM Model	Features	Feature Count	Loss	Rank on Kaggle
EfficientNet-B4	-	Full feature vectors	2048	0.07212	278
ResNeXt-101 32×8 d	-	Full feature vectors	2048	0.06259	191
ResNeXt-101 + BiLSTM	3×256	ResNeXt-101 predictions	6	0.05540	64
ResNeXt-101 + BiLSTM	3×256	Features with largest standard deviations	192	0.05212	37
ResNeXt-101 + BiLSTM	3×256	Features of weights with largest magnitude	192	0.05365	42
ResNeXt-101 + BiLSTM	3×256	Features of weights with smallest magnitude	192	0.05136	34
ResNeXt-101 + BiLSTM	3×256	Features of weights with smallest magnitude	120	0.05035	29
ResNeXt-101 + BiLSTM	2×128	PCA features	192	0.05207	36
ResNeXt-101 + BiLSTM	3×128	PCA features	192	0.05198	35
ResNeXt-101 + BiLSTM	3×256	PCA features	192	0.05096	30
ResNeXt-101 + BiLSTM	3×256	PCA features	120	0.05022	29
ResNeXt-101 + BiLSTM	3×256	PCA features + ResNeXt-101 predictions	120	**0.04989**	**27**

**Table 3 sensors-20-05611-t003:** Accuracy, ROC AUC, sensitivity and specificity on the validation set for our best model, namely a bidirectional LSTM trained on 120-dimensional feature vectors obtained by applying PCA on the 2048-dimensional features vectors produced by a ResNeXt-101 32×8d network. The same metrics are reported for the plain ResNeXt-101 32×8d model. Results are report for all intracranial hemorrhage subtypes.

ICH Type	ResNeXt-101 and Bidirectional LSTM							
	Accuracy		ROC AUC		Sensitivity		Specificity	
	on Slices	on Scans	on Slices	on Scans	on Slices	on Scans	on Slices	on Scans
Any	0.9600	0.9516	0.9843	0.9792	0.8605	0.9436	0.9781	0.9576
Epidural (EPH)	0.9970	0.9892	0.9851	0.9414	0.4430	0.4000	0.9988	0.9973
Intraparenchymal (IPH)	0.9832	0.9368	0.9927	0.9834	0.8142	0.8639	0.9927	0.9620
Intraventricular (IVH)	0.9920	0.9651	0.9970	0.9866	0.9102	0.9127	0.9948	0.9757
Subarachnoid (SAH)	0.9752	0.9220	0.9821	0.9609	0.6624	0.6944	0.9921	0.9767
Subdural (SDH)	0.9627	0.9140	0.9682	0.9451	0.6817	0.7203	0.9837	0.9601
	**ResNeXt-101**							
Any	0.9543	0.9409	0.9752	0.9782	0.7866	0.9436	0.9848	0.9388
Epidural (EPH)	0.9967	0.9879	0.9703	0.9479	0.2405	0.4000	0.9992	0.9959
Intraparenchymal (IPH)	0.9809	0.9449	0.9883	0.9806	0.6865	0.8377	0.9974	0.9819
Intraventricular (IVH)	0.9905	0.9570	0.9953	0.9889	0.8325	0.9127	0.9960	0.9660
Subarachnoid (SAH)	0.9697	0.9126	0.9644	0.9582	0.5024	0.7222	0.9949	0.9583
Subdural (SDH)	0.9604	0.9086	0.9576	0.9408	0.5993	0.8182	0.9873	0.9301

**Table 4 sensors-20-05611-t004:** Distribution of hemorrhage subtypes in the dataset of 100 scans that were held out for the subjective assessment by radiologists.

Label	EPH	IPH	IVH	SAH	SDH	Any (ICH)
Positive	0	29	16	25	32	52
Negative	100	71	84	75	68	48

**Table 5 sensors-20-05611-t005:** Performance metrics for annotations collected from three radiologist versus the predictions of the ResNeXt-101 32×8 d and our joint ResNeXt-101 32×8 d and bidirectional LSTM framework. To prevent cheating, the doctors did not see the ground-truth labels. Results are reported on 100 CT scans held out from the validation set.

	Doctor #1					
	EPH	IPH	IVH	SAH	SDH	Any
Accuracy	0.92	0.70	0.87	0.71	0.71	0.81
Sensitivity	-	1.0	0.50	0.04	0.62	0.98
Specifity	0.92	0.58	0.94	0.93	0.75	0.62
ROC-AUC	-	0.79	0.72	0.49	0.69	0.80
	**Doctor #2**					
	**EPH**	**IPH**	**IVH**	**SAH**	**SDH**	**Any**
Accuracy	0.97	0.89	0.95	0.91	0.94	0.95
Sensitivity	-	0.90	0.88	0.64	0.81	0.94
Specifity	0.97	0.89	0.96	1.00	1.00	0.94
ROC-AUC	-	0.89	0.92	0.82	0.91	0.95
	**Doctor #3**					
	**EPH**	**IPH**	**IVH**	**SAH**	**SDH**	**Any**
Accuracy	0.96	0.91	0.97	0.83	0.91	0.96
Sensitivity	-	0.90	0.94	0.40	0.81	0.96
Specifity	0.96	0.92	0.98	0.97	0.96	0.96
ROC-AUC	-	0.90	0.95	0.68	0.88	0.96
	**ResNeXt-101**					
	**EPH**	**IPH**	**IVH**	**SAH**	**SDH**	**Any**
Accuracy	1.00	0.90	0.94	0.86	0.90	0.98
Sensitivity	-	0.90	0.94	0.72	0.78	0.98
Specifity	1.00	0.90	0.94	0.91	0.96	0.98
ROC-AUC	-	0.90	0.94	0.81	0.87	0.98
	**ResNeXt-101 and bidirectional LSTM**					
	**EPH**	**IPH**	**IVH**	**SAH**	**SDH**	**Any**
Accuracy	1.00	0.92	0.95	0.83	0.93	0.97
Sensitivity	-	0.90	0.94	0.48	0.78	0.96
Specifity	1.00	0.93	0.95	0.95	1.00	0.98
ROC-AUC	-	0.91	0.94	0.71	0.89	0.97

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
