# Peer review of "Accurate and Efficient Intracranial Hemorrhage Detection and Subtype Classification in 3D CT Scans with Convolutional and Long Short-Term Memory Neural Networks"

_sensors, 2020, doi:10.3390/s20195611_

Round 1
Reviewer 1 Report
The article presents a deep learning approach for intracranial hemorrhage detection.
The novelty lies in combining CNNs with LSTMs integrating a feature selection method for more efficient processing. Also, in contrast to most of the prior work, the proposed network is able to classify subtypes of hemorrhages.
The article is written very well, discusses prior work and the proposed approach in detail.
The evaluation is done both on the RSNA Intracranial Hemorrhage Detection challenge and in comparison to experienced radiologists. Results show that the proposed methods achieves competitive outcomes despite its focus on efficiency.
Source code is provided at github in a public repository.
In my opinion, the article is of interest to the community, has no significant flaws and can be published in its current form.
Minor comments:
* Figure 4: EDH -> EPH
Author Response
We thank the reviewer for his comments and appreciation.
Reviewer 2 Report
The paper describes a system to detect intracranial hemorrhage in CT scans based on CNN and LSTM networks. The work is well written, and the approach is well grounded on both knowledges of neural network and the clinic in brain CT scans. Also, thanks for comparing the performance of the proposed network with annotation of clinical experts in Discussion section. References are adequate, and the authors state clearly the differences with their recent work in the same area and analysis of the proposed method in various angles. The paper can be accepted in its present form.Author Response
We thank the reviewer for his comments and appreciation.
Reviewer 3 Report
line 177, authors say that they train the CNN, then they refer to a pretrained architecture, did the author carry out the training of the CNN or not? Is it truly a CNN or a residual network? Is the input size conditioned by the selection of the pretrained architecture? Are you doing "transfer learning"?
When talking about windows, there is confussion between spatial and intensity windows, for instance, starting from line 200 please indicate HU intensity window, or use the work "interval" instead of window that has a more pronounced spatial meaning.
lines 242 - 250 describe the feature extraction network, I fail to understand the meaning of "cardinality" and "bottleneck".
line 271, it seems that you did only PCA, so please remove the comments about the softmax related feature selection, which BTW I do not understand.
line 276 I do not understand why you say that "reducing the FP and FN is possible by training on multiple slices..." I do not see the mathematics behind
line 285 how do you perform the two classification problems simultaneously? could it be you only have only one classification with an extra class "no hemorrhage"?
line 297 : you do not train the CNN, you use a pretrained network, or did I miss something?
figure 3, there is a classifier taking the output of the truncated pretrained network and generating some predictions labeled as resnet predictions. I do not understand because the resnet is not producing predictions on the hemorrhage class, it was trained for other kind of images. The BLSTM seems to be some kind of expansion system that converts vectors of 120 dimensions into 256 dimensions, which are later concatenated with the resnet predictions for a new classifier of model unknown, at least I can not find what these classifiers are.
after rereading, section 3.2 becomes more confuse. Too many generalities and too little specifics explaining the actual models. My impression is that the authors have first make some pytorch pipelines and later they try to explain it.
line 390 and table 3 increase my confusion, again, resnet was not trained for this problem, so it can not generate the hemorraghe labels.
additionally, it seems that there are two different validation sets, one in table 3 and 4, and a different one in section 5, please use different names.
conclusions: giving as conclusion the ranking in the kaggle challenge is not acceptable, BTW in the abstract there is no mention of the source of the data
Author Response
We would like to start by thanking the reviewer for reading our paper and raising useful comments that lead to a more clear presentation by eliminating ambiguities. We address each comment in the following.
line 177, authors say that they train the CNN, then they refer to a pretrained architecture, did the author carry out the training of the CNN or not? Is it truly a CNN or a residual network? Is the input size conditioned by the selection of the pretrained architecture? Are you doing "transfer learning"?
Author reply: As explained in lines 253-256, we take a pre-trained network and fine-tune it on our task. ResNeXt is a residual network, which is a particular type of CNN, i.e. as far as we know, residual networks represent a subtype of CNNs.
When talking about windows, there is confussion between spatial and intensity windows, for instance, starting from line 200 please indicate HU intensity window, or use the work "interval" instead of window that has a more pronounced spatial meaning.
Author reply: When we refer to "windows" in our paper, we always refer to intensity windows. We clarified in the paper that we discuss about "intensity windows" and not "spatial windows".
lines 242 - 250 describe the feature extraction network, I fail to understand the meaning of "cardinality" and "bottleneck".
Author reply: The terms "cardinality" and "bottleneck" are defined in [48], this being the paper that introduces the ResNeXt architecture. Xie et al. [48] define "cardinality" as the size of the set of transformations, and "bottleneck width" as the number of filters in the second convolutional layer in a ResNeXt block. We updated the paper accordingly to explain these details.
line 271, it seems that you did only PCA, so please remove the comments about the softmax related feature selection, which BTW I do not understand.
Author reply: In Table 2, we show results with various dimensionality reduction methods. If the reviewer looks at column "Features count", the reviewer should notice that models based on various dimensionality reduction methods have either 192 or 120 features (reduced from 2048). The last 5 methods are based on PCA, but the previous four are based on a different dimensionality reduction method which works as follows: Suppose the Softmax layer has 6 units (neurons), each assigning a vector of weights w to be applied on a vector of reduced features x from the previous layer. For each unit, we select the feature indexes using top(sort(abs(w)), 32). We concatenate the features indexes from the 6 units, obtaining a set of 192 features. Similarly, using top(sort(abs(w)), 20) will results in a total of 120 features.
line 276 I do not understand why you say that "reducing the FP and FN is possible by training on multiple slices..." I do not see the mathematics behind.
Author reply: Suppose we have the binary predictions for 10 CT slices (0, 0, 0, 1, 0, 1, 1, 0, 0, 0) and the ground-truth labels (0, 0, 0, 1, 1, 1, 1, 0, 0, 0). By considering neighboring slices, the 5th slice mislabeled as 0 (false negative) could be corrected by the system. The same could happen for a false positive example (consider the previous example with negated labels). We believe this example should explain our intuition, which is also confirmed by the experiments (please see Tables 2 and 3). In the paper, we completed our sentence by mentioning that "a spurious detection in one slice can be corrected by considering neighboring CT slices."
line 285 how do you perform the two classification problems simultaneously? could it be you only have only one classification with an extra class "no hemorrhage"?
Author reply: We consider 6 units (neurons), each performing binary classification with respect to the following classes: intracranial hemorrhage (ICH), epidural (EPH), intraparenchymal (IPH), intraventricular (IVH), subarachnoid (SAH), subdural (SDH). We note that an example x can have multiple positive labels. We use the multi-label log loss (see Eq. 1) to optimize our model. We provide more details in Section 3.3.
line 297 : you do not train the CNN, you use a pretrained network, or did I miss something?
Author reply: We take a pre-trained network and fine-tune / train it on our task.
figure 3, there is a classifier taking the output of the truncated pretrained network and generating some predictions labeled as resnet predictions. I do not understand because the resnet is not producing predictions on the hemorrhage class, it was trained for other kind of images. The BLSTM seems to be some kind of expansion system that converts vectors of 120 dimensions into 256 dimensions, which are later concatenated with the resnet predictions for a new classifier of model unknown, at least I can not find what these classifiers are.
Author reply: To fine-tune the network for our task (ICH detection and subtype classification), we removed the pre-trained classification layer and replaced it with our own classification layer (the classifiers are actually the Softmax layers being depicted separately in the figure). The whole architecture is fine-tuned starting from the pre-trained weights of the ResNeXt model. The ResNeXt model treats the slices 1,2, ..., N independently. The BiLSTM treats the slices jointly (an input example for the BiLSTM is a whole CT scan, which contains multiple slices). The classifiers are Softmax layers. We hope the above explanations make things more clear.
after rereading, section 3.2 becomes more confuse. Too many generalities and too little specifics explaining the actual models. My impression is that the authors have first make some pytorch pipelines and later they try to explain it.
Author reply: We added generic comments, as we believe that the "Sensors" journal has a more generic audience. In the revised version, we added several additional details to makes things more clear.
line 390 and table 3 increase my confusion, again, resnet was not trained for this problem, so it can not generate the hemorraghe labels.
Author reply: As explained earlier, we took a pre-trained ResNeXt and fine-tuned it on our task.
additionally, it seems that there are two different validation sets, one in table 3 and 4, and a different one in section 5, please use different names.
Author reply: We used a single validation set, with results presented in Table 3. The set presented in Table 4 is a held-out subset of the validation set, which we used in section 5. Table 4 presents the subset used in Section 5. We never refer to the subset of 100 scans used in Section 5 as "the validation set", hence we believe there is no confusion.
conclusions: giving as conclusion the ranking in the kaggle challenge is not acceptable, BTW in the abstract there is no mention of the source of the data
Author reply: We extended our conclusions and updated our abstract accordingly.
Round 2
Reviewer 3 Report
The authors have answered my comments.